# Research Evolution on the Impact of Agronomic Practices on Soil Health from 1996 to 2021: A Bibliometric Analysis

Mohamed Houssemeddine Sellami [1,2,3,*] and Fabio Terribile [3,4]

1   Institute for Mediterranean Agricultural and Forestry Systems, National Research Council of Italy, P.le Enrico Fermi 1, Portici, 80055 Naples, Italy
2   ARCA 2010, Contrada Varignano, 81, Acerra, 81100 Naples, Italy
3   Interdepartmental Research Centre on the "Earth Critical Zone", Università degli Studi di Napoli Federico II, Via Università 100, Portici, 80055 Naples, Italy; fabio.terribile@unina.it
4   Department of Agriculture, Università degli Studi di Napoli Federico II, Via Università 100, Portici, 80055 Naples, Italy
*   Correspondence: mohamed.sellami@isafom.cnr.it

**Abstract:** In the last two decades, there has been a significant shift in focus towards soil health by international institutions, organizations, and scholars. Recognizing the vital role of soil in sustaining agriculture, ecosystems, and mitigating climate change, there has been a concerted effort to study and understand soil health more comprehensively. In this study, a bibliometric analysis was performed in order to determine the research trend of the articles published in the Scopus database in the last 26 years on soil health experimental studies and agronomic practices conducted in field conditions on agricultural soils. It has been observed that, after 2013, there has been a significant increase in research articles on soil health, with the USA and India research institutions ranking as the most productive on this topic. There is an asymmetry in international cooperation among research institutions, as well as for scholars. In addition, the research topic is gradually shifting from the effects of soil management strategies, especially nutrient management, on soil organic carbon and yield to the study of the impact of soil management on biochemistry and microbiological soil activities and greenhouse gas emissions. Future research should focus into more integrated approaches to achieve soil indicators enabling to evaluate the impact of sustainable management practices (e.g., cropping practices) on soil health.

**Keywords:** bibliometric analysis; soil health; agronomic practices

## 1. Introduction

In the last decade, soil health (from now on abbreviated as SH) has become a key topic in policy and research agenda. This soil health emphasis can be seen as a response to the alarming, degraded state of soils in European Union (EU) and elsewhere and also a push towards innovative sustainable management practices.

EU gave a great contribution to this result by building a series of initiatives. For instance, the EU "Soil Health and Food" Mission (recently renamed as "A Soil Deal for Europe") delivered substantial funding to SH research [1,2]. In addition, the EU established the "new EU Soil Strategy" (COM/2021/699 final), which is already the legal instrument towards soil protection and sustainable soil management in all EU countries; thus, it has a very different formal status from the Soil Mission reporting.

Finally, a new EU soil law proposal has just been proposed to the EU Parliament under the name "Soil Monitoring and Resilience (Soil Monitoring Law)" (COM/2023/416 final). The proposal set the definition of soil health as follows "'soil health' means the physical, chemical, and biological condition of the soil determining its capacity to function as a vital living system and to provide ecosystem services". Thus, a healthy soil is evaluated in view of its provision in delivering a set of ecosystem services which require to be measurable.

Soils health and its delivery of ecosystem services play also a key role in Sustainable Development Goals (SDG) in terms of contributing to food production (SDG2: "zero hunger"), good health and wellbeing (SDG3), water quality (SDG6: "clean water and sanitation"), sustainable production (SDG12: "sustainable consumption and production"), carbon capture and greenhouse gas emission (SDG13: "climate action"), and soil health and biodiversity preservation (SDG15: "life on land").

In last year, there have been many disputes about whether we «should» or «should not» use the term Soil Health [3,4]. Actually, this is not completely new; for instance, similar discussion appeared on the term «Soil & Land Quality» before and after FAO established the definition (1974). For the sake of this specific contribution, we agree with Janzen et al.'s [5] proposal on the use of "soil health metaphor" which states that "as long as soil health helps unearth better ways of knowing and sustaining land, let us use it, honing and redefining it as we learn". Thus here, we claim that the use of "Soil Health" can be a great help in our joint effort to overcome the huge divide between us scientists, experts, and citizens.

One key item in implementing SH is the "how" to measure it. It is well established that SH indicators require to fulfil the following criteria [6,7]: (i) easy to measure; (ii) measurable with practical, rapid, and inexpensive measurement methods; (iii) sensitive to variations in management without being reflective of merely short-term variation; (iv) relevant to soil ecosystem services; and (v) informative for management. In general, soil health indicators are based on physical, chemical, or biological measures [8]. It is not surprising that a multitude of soil-health indicators have been proposed.

Moreover, to these soil indicators are affected by management practices. In fact, agronomic practices have a significant impact on soil health and thus on sustainable agriculture and environmental conservation [9,10]. More specifically it has been demonstrated that practices', such as crop rotation [11], conservation tillage [12], cover cropping [13], and nutrient management [14], impact on soil health.

Attempting to put some order in the entire matter, Jian et al. [15] analysed over 500 studies on soil health and quality on 354 geographic sites (42 countries). They found 42 SH indicators and 46 SH background indicators (e.g., climate, elevation, soil type).

Between them, it is important to highlight those indicators produced after large SH initiatives, between them the Comprehensive Assessment of Soil Health by the Cornell Framework [16], the National Soil Health Institute (www.soilhealthinstitute.org, accessed on 20 November 2022) [17] and the US Department of Agriculture [18]. The EU Soil Mission indeed proposed a list of SH indicators [1] including soil pollutants, % carbon, soil structure, soil biodiversity, soil nutrients, and soil water regimes, but currently no soil moisture depletion, thresholds, and scores are yet established. Most interestingly, the EU Soil Mission, in order to progress further in implementing SH concepts, promoted the joint work of land users and scientists in the so named "Living Labs" where an interdisciplinary approach is foreseen. The aim being to establish by 2030—in EU countries—an effective network of 100 living labs and lighthouses (https://www.soilmissionsupport.eu/ll-lh, on 20 November 2022) to co-create SH knowledge, test solutions, and demonstrate their value in real-life conditions. Being a new initiative, the value of this new end-user engagement approach is yet to be demonstrated.

Considering the complex scenario described above, it is not surprising that there is no unified approach to assess the soil health with indicators, thresholds, scores that can be determined by standard operational methods under practical conditions (e.g., considering costs and time constrains). In fact, different indicator systems are being used by institutions and by scientists with rather separate activities referring to various subdisciplines soil chemistry, soil physics, soil biology, and paedology, while at the same time there is a lack of unified approach. It seems fundamental to put some order to the entire matter which currently the SH research scenario looks rather chaotic with a very large number of different indicators working in different settings. Indeed, outstanding SH reviews have been produced, e.g., [3,19,20], but despite this evidence, the very large development of

many and diverse SH papers requires both update of reviews and in-depth analysis of SH indicators from different perspectives. This is crucial if current SH policies must be implemented. Indeed, policies require coherent operational approach to be profitably implemented also to large territories.

Considering this scenario and the key issue of SH indicators, in this specific contribution we aimed to produce some understanding on SH indicators by bibliometric analysis focused on SH research evolution in the period 1996–2021. We produced such analysis giving special emphasis on SH experimental studies and agronomic practices conducted in field condition on agricultural soils because we believe that an in-depth analysis must start from the analysing the large number of SH experimental work already produced in last decades as a preliminary step towards more comprehensive SH assessments. In addition, considering the current development of soil policies in the world (e.g., SDG, USA, EU) we have analysed these results also in view of countries, continents, research institutions, and scientists where SH research was produced.

The remaining sections of this work are organized as follows. Material and Methods are presented in the next main section, which details the search strategy used to gather the relevant literature, criteria for inclusion and exclusion of studies, and data extraction procedure. Then, the study issue is discussed in light of the major findings of the bibliometric analysis, such as publishing trends, authorship patterns, and co-occurrence author's keywords analysis.

## 2. Materials and Methods

### 2.1. Methodology

The search was conducted on January 2022, using a bibliometric analysis as a statistical tool to evaluate the scientific literature related to the field of study. A Bibliometric analysis is a quantitative research method that involves the systematic examination of scientific publications to uncover patterns, relationships, and trends within the scientific literature. By applying statistical techniques and data visualization tools, bibliometric analysis provides valuable insights into the structure and dynamics of the academic community, aiding researchers, institutions, and policymakers in decision making and strategic planning [21]. Despite its virtues, bibliometric analysis is still relatively new field in agricultural research. In the last 20 years (from 2003 to 2023), just 110 and 43 research publications on bibliometric analysis, were published in agronomy and soil science, respectively, according to the Web of Science database.

### 2.2. Data Source and Search Criteria

The bibliometric analysis performed in this study follows the protocol established by the Collaboration for Environmental Evidence (CEE) as defined by Pullin et al. [22]. There-fore, bibliometric analysis was performed using specific inclusion and exclusion criteria for selecting papers. Four selection criteria were evaluated for selecting papers for the bibliometric analysis: (I) studies conducted only under field conditions, but not under green-house conditions, pots, laboratory, and mesocosm; (II) studies that focused on agricultural soils and excluded grassland, forest, mining soils and urban soils; (III) studies included cropland and excluded potted plant, soilless culture, hydroponics, and aquaponic, and (IV) studies that focused on agronomic management except land cover, land use, cropping patterns, and integrated farming system. In this last selection, it was important to exclude both papers not oriented towards farm experiments but more generic towards landscape scenarios and papers involving farm fishing system.

The research question inquiry for this bibliometric analysis (How agricultural practices impact on soil health?) was formulated utilizing the PICOL (Population/Intervention, Comparator/Outcome and Location) model (Table 1), as outlined in the CEE protocol.

**Table 1.** The eligibility criteria in relation to research question key elements.

| PICOL | Description |
|---|---|
| Population | **Studies:** Study in open-field with experimental design and model and excluded greenhouse, laboratory and mesocosm |
| | **Soil:** Agricultural soils and excluded grassland, forest, mining soils and urban soils |
| | **Crops:** Study included cropland and excluded potted plant, soilless culture, hydroponics and aquaponic |
| Intervention | All agronomic management except land cover, land use, cropping patterns and integrated farming system |
| Comparator | Impacts and/or benefits |
| Outcome | Soil health indicators |
| Location | All the world |

The PICOL framework was used to incorporate the term of "soil health" into the title, abstract, and keywords of the literature gathered via the Scopus database and to identify articles that were written in English, between 1980 and 2021 in peer-reviewed journals and limited to these Scopus subject area: Agricultural and Biological Sciences, Environmental Science, Biochemistry, Genetics and Molecular Biology, Earth and Planetary Sciences, Immunology and Microbiology, and Multidisciplinary.

*2.3. Screening*

After performing the initial search, a total of 3327 articles were identified and were subsequently added to Endnote software (Version 20.2.1, Clarivate Analytics). Figure 1 describes the procedure for selecting articles. After applying the aforementioned inclusion criteria to select titles, abstracts, and full-text studies, a total of 2342 articles were excluded. As a consequence, only 985 documents fulfilled the eligibility criteria and were considered for the bibliometric analysis.

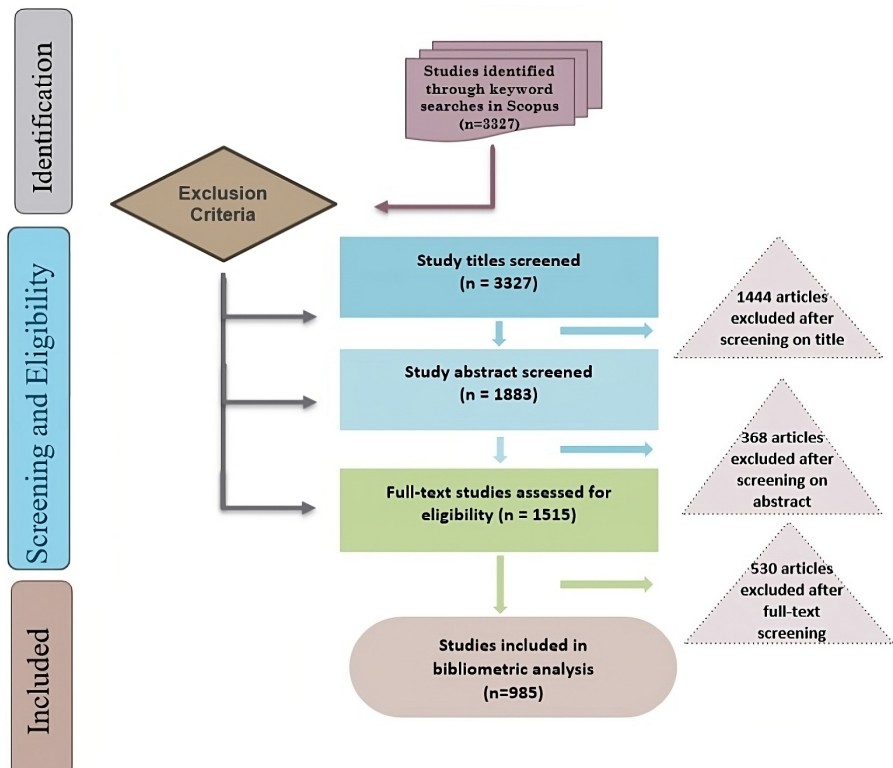

**Figure 1.** Prisma flowchart depicting the article selection procedure (n represent the number of studies).

*2.4. Data Analysis*

This study aims to analyse the research evolution of 985 scientific papers focusing on the influence of agronomic practices on soil health over a 26-year period (1996–2021). To achieve this objective, various bibliometric indicators are applied including the number of publications per scholar, number of citations, number of countries, number of institutions and journals, H-index, the journal impact factors as well as keyword co-occurrence analysis across different time periods.

The quantitative analysis was carried out using SciVal data platform, developed by Elsevier, a tool that evaluates the research performance, identify collaboration opportunities, and discover emerging trends within their field based on the Scopus database [23,24]. To create a network of co-occurrence author's keywords, we used VOSviewer [25].

## 3. Results and Discussion

*3.1. Evolution of Scientific Production*

A comprehensive selection process was conducted to identify 985 research papers pertaining to soil health in agricultural soils. These papers were chosen based on pre-determined eligibility, inclusion, and exclusion criteria for the purpose of conducting a bibliometric analysis. The Table 2 shows the evolution of scientific experimental papers published on soil health under agricultural soils since 1996 until 2021. The analysis of the key attributes of scholarly articles pertaining to soil health in this table is conducted in five distinct periods, each encompassing a duration of five years, except for the final period which spans six years.

**Table 2.** Characteristics of scientific production from 1996 to 2021.

| Period | Articles | Authors | Coutries | TC | TC/A | Journals |
|---|---|---|---|---|---|---|
| **1996–2000** | 7 | 22 | 5 | 410 | 58.6 | 5 |
| **2001–2005** | 26 | 110 | 13 | 1386 | 53.3 | 22 |
| **2006–2010** | 53 | 220 | 16 | 2993 | 56.5 | 38 |
| **2011–2015** | 162 | 681 | 29 | 4145 | 25.6 | 83 |
| **2016–2021** | 737 | 3151 | 66 | 12,229 | 16.6 | 209 |

TC: Total citations on 25 June 2023; TC/A: Citations per article.

The data reveal a noticeable increase in research interest regarding soil health in agricultural soils over the analysed period. This trend is particularly evident in the last six years (2016–2021), during which more than 70% and 50% of the total scientific publications and total citations on this topic were published and cited, respectively. It is observed that there has been a more than 100-fold and 30-fold increase in the number of articles and total citations during the recent period, respectively, in comparison to the initial five-year period (1996–2000). The impetus behind this could be after the United Nations General Assembly, at its sixty-eighth session on 20 December 2013, declared 2015 as the International Year of Soils (A/C.2/68/L.21). The scientific production in this research topic experienced a good annual growth rate of 24.26%, with a clear expansional trend of scientific production since 2011 until 2021 (Figure 2). However, in the 2010s, the importance of soil health has been recognized in various international frameworks, such as the Sustainable Development Goals (SDGs), particularly Goal 15: Life on Land. The goal includes a target (15.3) to achieve a land degradation-neutral world by 2030, highlighting the importance of healthy soils for sustainable development [26]. The number of published research papers has fluctuated over the last 26-years, reaching a peak of 246 during 2021, where the total citations peaking at 2662 in 2019 (data not shown).

In spite of the notable expansion in the quantity of articles and citations, there has been a contrasting trend observed in the number of citations per document. Over the past six years, there has been a significant decline of 70% in the citations received per article, in comparison to the initial years (Table 2). The decline in the number of citations received per article over time can be attributed to several factors such as increased competition

between scholars. When the number of published articles continues to grow, the pool of potential citations for each article becomes larger. With more research being conducted and published, it becomes increasingly challenging for any single article to stand out and receive a high number of citations. Moreover, over time, researchers have become increasingly specialized in their fields, which may lead to a narrower focus and a decrease in the number of articles they cite. This increased specialization can result in smaller, more focused citation networks. It's important to note that while the number of citations per article may decline, it does not necessarily indicate a decrease in the quality or impact of the research. It may simply be a reflection of the changing landscape of scholarly communication and the growth of the research community.

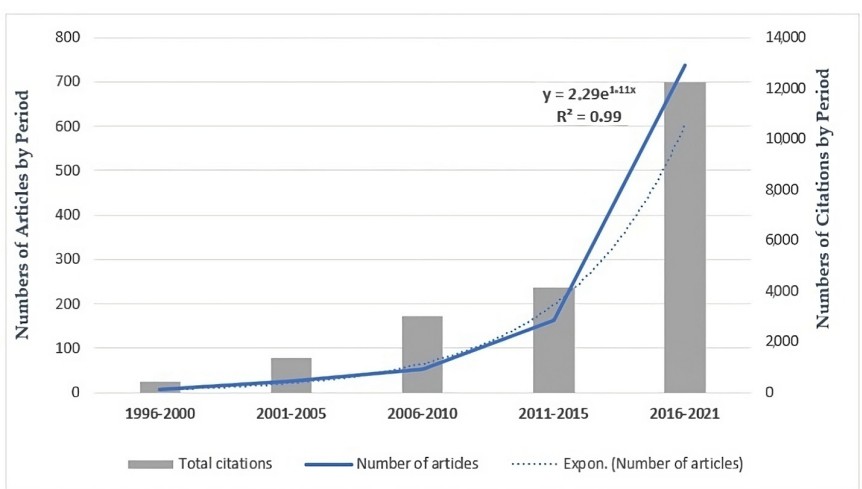

**Figure 2.** Evolution of the number of articles, total citations, and exponential variation between periods. The dot line refers to the calculated exponential curve fitting the data (number of articles).

The observed increase is noteworthy in relation to various bibliometric indicators, including the number of authors. In the initial period (1996–2000), there were 22 authors, with an average of 3 authors per article. However, in the most recent six-year period, the number of authors has increased 143 times (Table 2). It is worth noting that despite this significant growth, there has been no significant variation in the average number of authors per article (4 authors per article).

It can be noted that in the first five years, articles from only five countries were submitted, but that number has increased more than 13 times in the last six years.

On the other hand, the documents were published in 250 distinct sources. In the initial period, the articles were published in a total of five journals. However, during the subsequent period from 2011 to 2015, there was a substantial increase in the number of journals, reaching a total of 83. Over the course of the last six years, this number further rose to 209 journals.

Most of the bibliometric indicators mentioned above show a very considerable increase in the relevance of this field of study over the previous ten years, further highlighting the strength of the current trend in this line of research.

### 3.2. Analysis of Scientific Production

#### 3.2.1. Subject Area and Journals

Over a span of 26 years (1996–2021), a comprehensive analysis was conducted on 985 research articles that investigated the impact of agronomic practices on soil health in agricultural soils. These articles were classified into 22 distinct subject areas as per the Scopus database. Figure 3 shows that 51.9% of research articles were placed in the subject area of Agricultural and Biological Sciences, while 21.3% were classified in the Environmental Science category. Afterwards, the most important categories are Earth

and Planetary Sciences (6.1%), Immunology and Microbiology (4.7%), and Biochemistry, Genetics, and Molecular Biology (3.5%).

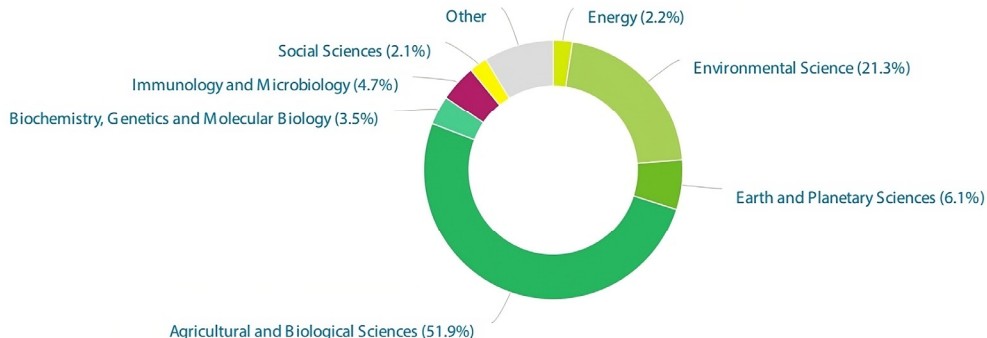

**Figure 3.** Donut chart of relative publication per Scopus subject Area. Source: SciVal.

Therefore, across the time period studied (1996–2021), Figure 4 illustrates the evolution of the seven key subject areas as Scopus links articles on the research topic. Only four thematic areas (Agricultural and Biological Sciences, Environmental Science, Earth and Planetary Sciences, and Immunology and Microbiology) had articles published during the whole 26-year period under consideration. Agriculture, Biology, and Environmental aspects were believed to be the most pertinent in the analysis of the effect of soil management on soil health, although Microbiology, Biochemistry, Genetic and Molecular biology cannot be overlooked in soil health.

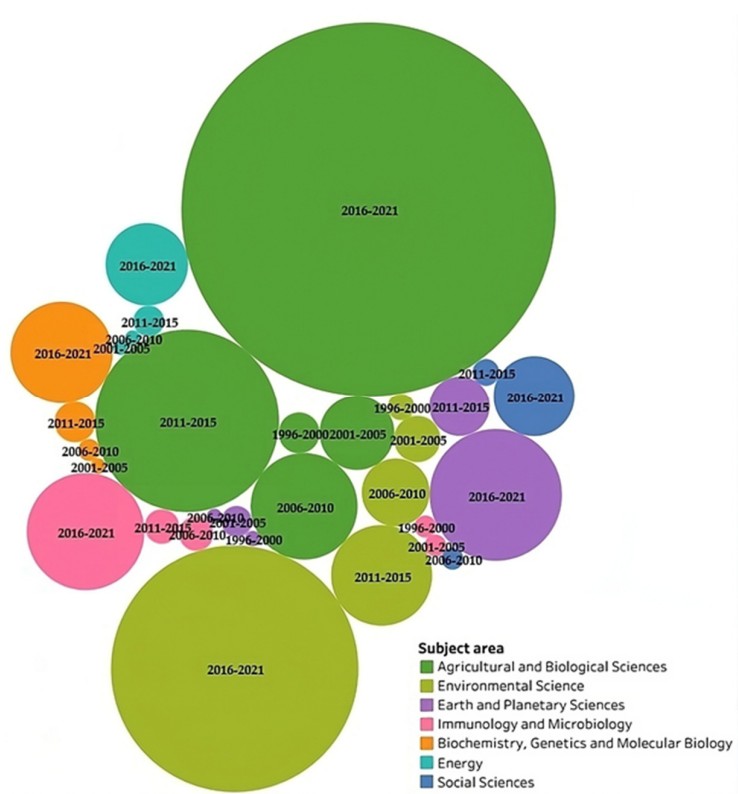

**Figure 4.** Packed bubbles of the growth trend of the main Scopus subject Area per period. Source: Tableau.

Table 3 shows the most 20 prolific journals for the number of articles on the research topic. A total of 60% of these top 20 journals with the highest scientific production are found in the first quartile (Q1) of CiteScore index in 2022 with impact factor ranging from

2.4 (Archives of Agronomy and Soil science and Journal of Environmental Quality) to 9.7 (Soil Biology and Biochemistry). On the other hand, 55% of this group of journals are located in Europe (mostly Netherlands), 25% in North America (USA), 15% from Asia (India) and the last 5% come from Australia. Since 1996 until 2021, a total of 985 research articles have been published across 250 journals. Notably, the most 20 prolific journals receiving 416 articles, which accounts for 42% of the overall scientific production. During the time frame of our investigation, we identified two journals, Applied Soil Ecology and Soil and Tillage Research, which together accounted for 9.54% and 20.51% of all articles published and overall Total citations, respectively. Moreover, these two journals are considered among journals with high number of citations per article with 41.4 and 51.1 citations per article for Applied Soil Ecology and Soil and Tillage Research, respectively. Moreover, they have the highest H index for articles published on soil health with 24 and 27 for Applied Soil Ecology and Soil and Tillage Research, respectively. Therefore, during the last 6 years analysed (1996–2000), 307 articles on the research topic were published in the 20 most prolific journals, but the five articles on this topic were spread among 5 prolific journals over the first 5 years (1996–2000). In addition, as seen by the increase in research papers and the broad range of journals, impact of soil management on soil health has become an attractive subject for more journals and authors over time.

### 3.2.2. Most Productive Authors

Using bibliometric metric indicators such as number of publications, total citations, and average citations per article, Table 4 lists the 10 most productive authors on soil health in our research topic. The ten authors in question have published 101 articles and have been cited 2348 times, accounting for 10.25% and 11.09% of total publications and citations, respectively.

It is noteworthy that authors of Asian origin present higher productivity in the field of research, with India (70%) and Pakistan (10%), being particularly prominent. North America, specifically the United States, accounted for 20% of the represented authors.

The most productive author is Das, Anup, from ICAR Research Complex for NEH Region, with a total of 16 articles published during 14 years (2008–2021) and receiving a total of 255 citations, who has the highest H-index of 11, followed by Jat, Mangai Lal with a total of 12 research articles, who is also the author with the highest number of citations and the highest number of citations per articles, with a total of 778 and 64.83, respectively.

Finally, two Asian authors, Farooq, Muhammad A. from University of Agriculture, Faisalabad and Jat, Mangi Lal from International Maize & Wheat Improvement Center, showed a higher percentage of international co-authorship during the period analysed (1996–2021), with 87.5% and 50.0%, respectively (Table 5). The prominence of Asian authors among the top 10 most relevant contributors is evident, a trend that is likely reflected in the numbers and supported by bibliometric analysis. This evidence highlights the importance of soil health research, especially in Indian Research Institutes. However, it is crucial to maintain perspective regarding their publications, primarily confined to local journals and limited international impact. This serves as a reminder that while numerical metrics highlight certain patterns, considering the qualitative aspects of influence is equally vital.

### 3.2.3. Most Productive Countries and Affiliations

Only 37.44% of the nations across the globe (73 countries) have contributed to this particular field of research. Figure 5 shows that 48.34% of research articles are found in Asia (642 publications), followed by North America (338 publications, 25.45%), and Europe (174 publications, 13.10%). These three continents together represent 87% of papers published in the soil health research field. In contrast, our bibliometric analysis reveals that Oceania, Africa and South America exhibit the lowest number of publications with 75, 51, and 48 articles, respectively.

**Table 3.** Top 20 prolific journals (1996–2021).

| Journal | A | TC | TC/A | Hi (A) | Hi (J) | IF (J) | CiteScore 2022 | C | FA | LA | R (A) | | | | |
|---|---|---|---|---|---|---|---|---|---|---|---|---|---|---|---|
| | | | | | | | | | | | 1996–2000 | 2001–2005 | 2006–2010 | 2011–2015 | 2016–2021 |
| APPLIED SOIL ECOLOGY | 48 | 1988 | 41.4 | 24 | 136 | 4.8 | 8.7 (Q1) | Netherlands | 1999 | 2021 | 1 (2) | 1 (3) | 1 (8) | 4 (7) | 3 (28) |
| SOIL AND TILLAGE RESEARCH | 46 | 2352 | 51.1 | 27 | 162 | 6.5 | 12.7 (Q1) | Netherlands | 1998 | 2021 | 5 (1) | 3 (2) | 36 (1) | 3 (10) | 1 (32) |
| COMMUNICATIONS IN SOIL SCIENCE AND PLANT ANALYSIS | 35 | 463 | 13.2 | 11 | 75 | 1.8 | 3.0 (Q2) | USA | 2005 | 2021 | - | 10 (1) | - | 1 (14) | 7 (20) |
| INDIAN JOURNAL OF AGRICULTURAL SCIENCES | 32 | 168 | 5.3 | 8 | 30 | 0.4 | 0.9 (Q4) | India | 2004 | 2021 | - | 2 (2) | 6 (2) | 5 (7) | 6 (21) |
| AGRONOMY | 30 | 270 | 9.0 | 11 | 67 | 3.7 | 5.2 (Q1) | Switzerland | 2019 | 2021 | - | - | - | - | 2 (30) |
| SOIL SCIENCE SOCIETY OF AMERICA JOURNAL | 26 | 845 | 32.5 | 13 | 184 | 2.9 | 4.9 (Q2) | USA | 2005 | 2021 | - | 21 (1) | 35 (1) | 77 (1) | 5 (23) |
| AGRONOMY JOURNAL | 23 | 186 | 8.1 | 7 | 145 | 2.1 | 4.3 (Q2) | USA | 2017 | 2021 | - | - | - | - | 4 (23) |
| INDIAN JOURNAL OF AGRONOMY | 18 | 81 | 4.5 | 5 | 25 | 0.036 | 0.5 (Q4) | India | 2008 | 2019 | - | - | 7 (2) | 2 (10) | 31 (6) |
| GEODERMA | 17 | 766 | 45.1 | 14 | 190 | 6.1 | 12.9 (Q1) | Netherlands | 2004 | 2021 | - | 13 (1) | - | - | 8 (16) |
| AGRICULTURE, ECOSYSTEMS AND ENVIRONMENT | 16 | 788 | 49.3 | 13 | 200 | 6.6 | 10.2 (Q1) | Netherlands | 2000 | 2020 | 3 (1) | - | 10 (1) | 8 (3) | 12 (11) |
| ARCHIVES OF AGRONOMY AND SOIL SCIENCE | 16 | 245 | 15.3 | 9 | 49 | 2.4 | 5.5 (Q1) | United Kingdom | 1998 | 2021 | 4 (1) | 5 (1) | 12 (1) | 9 (3) | 14 (10) |
| SUSTAINABILITY (SWITZERLAND) | 15 | 129 | 8.6 | 7 | 136 | 3.9 | 5.8 (Q1) | Switzerland | 2018 | 2021 | - | - | - | - | 9 (15) |
| FRONTIERS IN MICROBIOLOGY | 13 | 437 | 33.6 | 9 | 201 | 5.2 | 7.8 (Q1) | Switzerland | 2016 | 2021 | - | - | - | - | 10 (13) |
| SCIENTIFIC REPORTS | 13 | 428 | 32,9 | 12 | 282 | 4.6 | 7.5 (Q1) | United Kingdom | 2016 | 2021 | - | - | - | - | 11 (13) |
| JOURNAL OF ENVIRONMENTAL QUALITY | 12 | 116 | 9.7 | 6 | 183 | 2.4 | 6.6 (Q1) | USA | 2012 | 2021 | - | - | - | 63 (1) | 13 (11) |
| SOIL BIOLOGY AND BIOCHEMISTRY | 12 | 588 | 49.0 | 11 | 250 | 9.7 | 14.3 (Q1) | United Kingdom | 2004 | 2021 | - | 20 (1) | 34 (1) | 31 (2) | 21 (8) |
| SOIL RESEARCH | 11 | 142 | 12.9 | 5 | 92 | 1.6 | 3.6 (Q2) | Australia | 2014 | 2021 | - | - | - | 32 (2) | 17 (9) |
| EUROPEAN JOURNAL OF SOIL BIOLOGY | 11 | 422 | 38.4 | 8 | 84 | 4.2 | 5.9 (Q1) | France | 2008 | 2021 | - | - | 4 (2) | 18 (2) | 24 (7) |
| HORTSCIENCE | 11 | 167 | 15.2 | 7 | 100 | 1.9 | 3.2 (Q2) | USA | 2007 | 2020 | - | - | 5 (2) | 6 (5) | 48 (4) |
| JOURNAL OF ENVIRONMENTAL BIOLOGY | 11 | 61 | 5.5 | 5 | 57 | 0.7 | 1.4 (Q3) | India | 2013 | 2021 | - | - | - | 7 (4) | 25 (7) |

(A): number of articles; (TC): number of citations; (TC/A): number of citations per article; Hi (A): h-index of articles; Hi (J): h-index of journal; (C): Country; (FA): First article; (LA): Last article; (Q): Quartile; (R): Rank; IF (J): Impact factor (IF) produced by InCites Journal of Citation Reports (www.clarivate.com) on 6 July 2023.

**Table 4.** Top 10 most relevant authors on soil health from 1996 to 2021.

| Authors | A | TC | TC/A | Institution | C | FA | LA | H Index |
|---------|---|----|------|-------------|---|----|----|---------|
| Das, Anup | 16 | 255 | 15.94 | ICAR Research Complex for NEH Region | India | 2008 | 2021 | 11 |
| Jat, Mangi Lal | 12 | 778 | 64.83 | International Maize & Wheat Improvement Center (CIMMYT) | India | 2004 | 2020 | 10 |
| Babu, Subhash | 11 | 136 | 12.36 | ICAR Research Complex for NEH Region | India | 2013 | 2021 | 8 |
| Sainju, Upendra M. | 11 | 117 | 10.64 | United States Department of Agriculture (USDA) | USA | 2018 | 2021 | 5 |
| Ghimire, Rajan P. | 10 | 131 | 13.10 | New Mexico State University | USA | 2019 | 2021 | 5 |
| Yadav, Gulab Singh | 9 | 168 | 18.67 | ICAR Research Complex for NEH Region | India | 2013 | 2021 | 7 |
| Dwivedi, Brahma S. | 8 | 252 | 31.50 | ICAR—Indian Agricultural Research Institute | India | 2003 | 2020 | 7 |
| Farooq, Muhammad A. | 8 | 135 | 16.88 | University of Agriculture, Faisalabad | Pakistan | 2017 | 2021 | 5 |
| Singh, Vijendra K. | 8 | 242 | 30.25 | Central Research Institute for Dryland Agriculture India | India | 2003 | 2021 | 8 |
| Kumar, Sandeep | 8 | 134 | 16.75 | Indian Institute of Technology Guwahati | India | 2018 | 2021 | 6 |

(A): number of articles; (TC): number of citations; (TC/A): number of citations per article; (C): Country; (FA): First article; (LA): Last article.

**Table 5.** International co-authorship of top 10 most relevant authors on soil health from 1996 to 2021.

| Authors | C | IC | IC (%) | Cited Publications |
|---------|---|-----|--------|---------------------|
| Das, Anup | India | 3 | 18.8 | 16 |
| Jat, Mangi Lal | India | 6 | 50.0 | 12 |
| Babu, Subhash | India | 1 | 9.1 | 11 |
| Sainju, Upendra M. | USA | 3 | 27.3 | 8 |
| Ghimire, Rajan P. | USA | 1 | 10.0 | 8 |
| Yadav, Gulab Singh | India | 2 | 22.2 | 9 |
| Dwivedi, Brahma S. | India | 1 | 12.5 | 8 |
| Farooq, Muhammad A. | Pakistan | 7 | 87.5 | 6 |
| Singh, Vijendra K. | India | 2 | 25.0 | 8 |
| Kumar, Sandeep | India | 1 | 12.5 | 8 |

(C): Country; (IC): International collaboration.

There is a growing recognition of the importance of soil health in policy and research arenas, and many countries and organizations have adopted soil health targets and strategies. For example, in Asia, soil degradation and nutrient depletion are major concerns due to intensive agriculture, population pressure, and climate change. The Food and Agriculture Organization (FAO) has identified several hotspots of soil degradation in Asia, including the Indus–Ganges basin, The Mekong Delta, and the Yangtze River Basin [27,28]. To address these issues, governments and research institutions in the region are investing in soil conservation and management practices, such as no-till farming, crop rotation, and agroforestry [29–31]. On the other hand, in North America, soil health has become a major focus for sustainable agriculture and conservation efforts in recent years. The USDA's Natural Resources Conservation Service (NRCS) has launched a Soil Health Partnership to promote the adoption of soil health practices, such as cover cropping, reduced tillage, and nutrient management [32]. There is also a growing body of research on the impacts of soil health on crop yields, carbon sequestration, and ecosystem services.

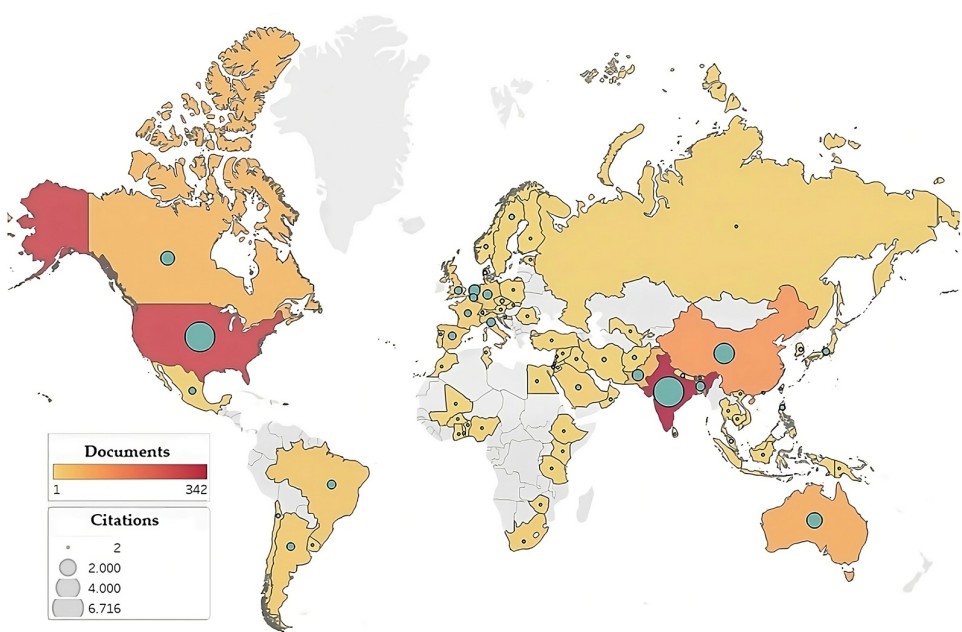

**Figure 5.** World map displaying the distribution of publications and total citations by country. Source: Tableau.

In Europe, soil health is a key priority under the EU's Common Agricultural Policy (CAP) and the European Green Deal. The EU has set targets for improving soil organic matter content, reducing erosion, and promoting biodiversity in agricultural landscapes. There are also several research networks and initiatives focused on soil health, such as the Soil Care project (https://www.soilcare-project.eu/, accessed on 20 July 2023) and the European Soil Partnership (https://www.europeansoilpartnership.org/, accessed on 20 July 2023).

Table 6 shows the most 10 prolific countries in terms of number of publications, total citations, and number of citations per article. India has the highest number of articles published and cited on the research topic, with a total of 342 articles and 6716 citations. The United States follows closely with 279 articles and 6506 citations, while China ranks third with 114 articles and 3050 citations. Additionally, it is noteworthy that India, the United States, and China possess the highest H-index values in relation to articles published on soil health, with respective scores of 41, 40, and 31. This observation suggests that these three countries play a prominent role in driving scientific research related to soil health. These three countries collectively accounted for 55% of the total published articles.

On the other hand, it is worth noting that Canada and Bangladesh are ranked first in terms of the number of citations per article, with a rate of 27.3. However, during each of the time periods analysed, India, the USA, and Australia consistently published scientific articles. Since the third 5 year period (2006–2010) under consideration, both China and the United Kingdom have made contributions to this research field, but Bangladesh did not do so until after 2011.

Based on the data presented in Table 7, it can be observed that Germany exhibits the highest level of international collaboration, accounting for 92.59% of its articles (25 out of 27) being written in collaboration. The United Kingdom follows closely with 81.82% (18 articles), while Bangladesh demonstrates a collaboration rate of 80% (20 articles). Brazil and Australia also engage in international collaboration, with rates of 72% (18 articles) and 60.81% (45 articles), respectively.

**Table 6.** Top 10 most productive countries on soil health from 1996 to 2021.

| Country | A | TC | TC/A | H Index | R (A) | | | | |
|---|---|---|---|---|---|---|---|---|---|
| | | | | | 1996–2000 | 2001–2005 | 2006–2010 | 2011–2015 | 2016–2021 |
| India | 342 | 6716 | 19.6 | 41 | 5 (1) | 1 (11) | 1 (26) | 1 (92) | 2 (212) |
| United States | 279 | 6506 | 23.3 | 40 | 1 (3) | 2 (6) | 2 (10) | 2 (21) | 1 (239) |
| China | 114 | 3050 | 26.8 | 31 | - | - | 3 (5) | 7 (5) | 3 (104) |
| Australia | 74 | 1747 | 23.6 | 24 | 2 (2) | 3 (2) | 5 (3) | 3 (13) | 4 (54) |
| Pakistan | 60 | 955 | 15.9 | 17 | - | 11 (1) | 14 (1) | 5 (8) | 5 (50) |
| Canada | 52 | 1422 | 27.3 | 19 | - | 4 (2) | 6 (3) | 4 (8) | 6 (39) |
| Germany | 27 | 647 | 24.0 | 13 | - | 5 (2) | - | 8 (4) | 8 (21) |
| Bangladesh | 25 | 682 | 27.3 | 15 | - | - | - | 6 (5) | 9 (20) |
| Brazil | 25 | 488 | 19.5 | 13 | - | 6 (1) | 10 (1) | 13 (2) | 7 (21) |
| United Kingdom | 22 | 450 | 20.5 | 13 | - | - | 16 (1) | 12 (3) | 11 (18) |

(A): number of articles; (TC): number of citations; (TC/A): number of citations per article; R: Rank.

**Table 7.** Top 10 most productive countries and international collaboration from 1996 to 2021.

| Country | NC | Main Collaborators | IC(%) | TC/A | |
|---|---|---|---|---|---|
| | | | | IC | NIC |
| India | 24 | United States, Australia, China, Canada, Germany | 14.91 | 34.5 | 17.1 |
| United States | 34 | India, Pakistan, China, Australia, Canada | 26.88 | 28.8 | 21.3 |
| China | 32 | India, United States, Pakistan, Australia, Canada | 53.51 | 29.0 | 24.1 |
| Australia | 26 | United States, India, China, Pakistan, Canada | 60.81 | 25.6 | 20.6 |
| Pakistan | 19 | United States, China, Australia, Canda, Germany | 53.33 | 22.0 | 8.9 |
| Canada | 10 | United States, India, China, Australia, Pakistan | 40.38 | 23.3 | 30.1 |
| Germany | 28 | United States, India, China, Australia, Pakistan | 92.59 | 22.7 | 39.5 |
| Brazil | 15 | United States, India, Australia, Canda, Germany | 72.00 | 19.8 | 18.7 |
| Bangladesh | 15 | India, China, Australia, United Kingdom, Netherlands | 80.00 | 25.9 | 33.0 |
| United Kingdom | 19 | United States, India, China, Australia, Pakistan | 81.82 | 23.6 | 6.5 |

NC = number of collaborations; IC(%) = percentage of articles made with international collaboration; TC/A: number of citations per article; IC: International collaboration; NIC: no international collaboration.

At only 14.91%, India has the lowest percentage of international cooperation. It should be noted that, with the exception of Canada, Germany, and Bangladesh, all countries included in the list of the most 10 productive countries exhibit a higher number of citations for articles produced through international collaboration compared to those produced without collaboration. Scientific collaboration is a reaction to the growing professionalization of science as noted by Beaver and Rosen [33]. As a result, international co-authored articles receive more citations than domestically co-authored articles because they receive more citations overall [34,35].

Figure 6 illustrates the international cooperation network among major countries, which is established through co-authorship analysis. According to Figure 6, the number of countries engaged in international collaboration related to the research topic is limited to 70. The countries have been grouped into seven clusters based on their specific fields of collaboration. These nations are categorized into seven clusters based on their areas of cooperation. Eighteen nations made up the first cluster (red), which was led by India. This cluster, which also included two of the top ten prolific countries namely Pakistan and Bangladesh. There was a total of 38% published articles coming from this cluster. The second cluster (green) is led by South Africa and includes 15 countries with 80 articles, which represents 6% of the total number of published articles, and included countries such as France, Belgium, and Mexico. The third cluster in blue, with a total of nine countries and it is led by Germany, with 86 articles (6.5% of the total articles). This group included Spain, Argentina, and Denmark. The fourth cluster (yellow), is led by Brazil and represent 4.97% of the total. This cluster included United Kingdom, in addition to Switzerland, Indonesia, Poland and Iraq. The fifth cluster (purple), is led by China with 250 articles (18.83% of the

total articles). The fifth cluster included Australia and Canada. The sixth cluster in Sky blue, is led by Italy and include Netherlands, Russian Federation and Mali. This cluster represents 3.46% of published articles. Finally, the seventh cluster (Orange) is led by the United States and represents the 21.76% of published articles on research topic during the last 26 years (1996–2021). This cluster included Nigeria, Jordan, and Sri Lanka.

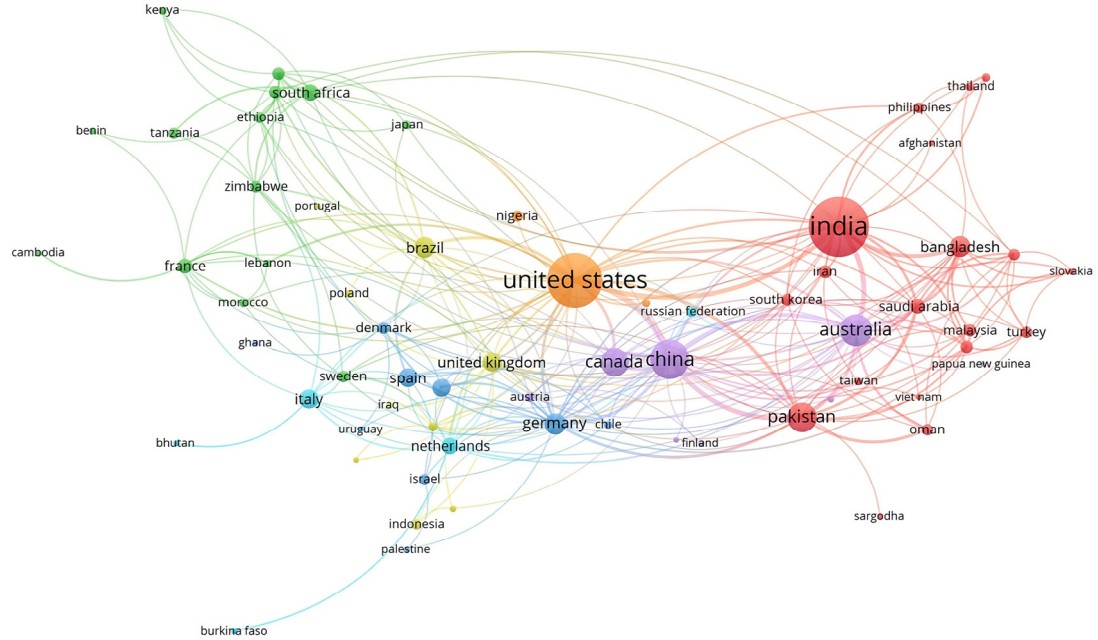

**Figure 6.** Country collaboration network visualization map. Colour: represents a cluster of country collaboration in the soil health research field; Nodes: represent countries (node size based on number of publications); Links: represent collaboration between two countries. Source: VOSviewer (clusters resolution 0.5; minimum cluster size 1 and no merge small clusters).

Our research topic on the impact of soil management on soil health under agricultural soils during the last 26 years (1996–2021) produced by 250 different affiliations. According the Table 8, the United States Department of Agriculture (USDA) form the United States is the most prolific institutions with 88 published articles and 2025 citations in this research topic. This institution shares the highest H-index of 24 with ICAR—Indian Agricultural Research Institute from New Delhi, India. In addition, 12.5% of articles from USDA were published with international co-authorship.

The nation with the greatest presence in this ranking is India, with five affiliations. Among the institutions considered, ICAR -Indian Agricultural Research Institute from New Delhi is the institution with the second highest number of articles of 71, which 19.7% from international co-authorship. In addition, this institution has the second most citations with 1612 total citations. The other four Indian institutions exhibit varying proportions of international co-authorship. For instance, Indian Council of Agricultural Research and ICAR—Indian Institute of Soil Science; Bhopal have published more than a third of their articles with international co-authorship. Conversely, ICAR—Research Complex for North Eastern Hill Region in Umiam and Punjab Agricultural University have the lowest percentage of international co-authorship, with 9.5 and 5.6%, respectively. The majority of articles having international co-authorship (more than 60%) are published by the Chinese Academy of Sciences and the University of Agriculture in Faisalabad, Pakistan.

**Table 8.** Top 10 most productive affiliations on soil health from 1996 to 2021.

| Institution | C | A | TC | TC/A | H Index | IC(%) | TC/A | |
|---|---|---|---|---|---|---|---|---|
| | | | | | | | IC | NIC |
| United States Department of Agriculture (USDA) | USA | 88 | 2025 | 23.01 | 24 | 12.5 | 21.1 | 23.3 |
| ICAR—Indian Agricultural Research Institute, New Delhi | India | 71 | 1612 | 22.70 | 24 | 19.7 | 41.0 | 18.2 |
| Indian Council of Agricultural Research | India | 60 | 1278 | 21.30 | 21 | 30.0 | 38.6 | 13.9 |
| ICAR—Research Complex for North Eastern Hill Region; Umiam | India | 31 | 379 | 12.23 | 12 | 9.7 | 33.0 | 10.0 |
| Agriculture and Agri-Food Canada | Canada | 29 | 558 | 19.24 | 14 | 27.6 | 24.3 | 17.3 |
| University of Agriculture; Faisalabad | Pakistan | 28 | 483 | 17.25 | 13 | 64.3 | 23.8 | 5.4 |
| Chinese Academy of Sciences | China | 22 | 593 | 26.95 | 11 | 68.2 | 31.1 | 18.0 |
| Cornell University | USA | 21 | 534 | 25.43 | 12 | 28.6 | 11.5 | 31.0 |
| Punjab Agricultural University | India | 18 | 261 | 14.50 | 7 | 5.6 | 2.0 | 15.2 |
| ICAR—Indian Institute of Soil Science; Bhopal | India | 18 | 579 | 32.17 | 12 | 33.3 | 30.3 | 33.1 |

(A): number of articles; (TC): number of citations; (C): Country; IC (%) = percentage of articles made with international collaboration; TC/A: number of citations per article; IC: International collaboration; NIC: no international collaboration.

It is worth mentioning that, apart from two institutions based in the United States and two institutions based in India, all affiliations listed among the top 10 most productive institutions demonstrate a greater number of citations per articles resulting from international collaboration relative to those produced without collaboration.

### 3.3. Keywords Co-Occurrence Network Analysis

A total of 985 articles containing 2460 author keywords were taken into account for this study. The 20 most frequently used author keywords in the 985 research articles related to soil health during the period 1996–2021 are shown in Table 8. We expected soil health to be one of the most prominent keywords on the list (240 articles, 24.4% of total articles), as it was one of the most frequently searched terms. In the first five years of the analysis (1996–2000), this keyword occurred in five articles. It peaked in the last six years (2016–2021), when 169 documents included this term. On the other hand, it is interesting that we additionally identified yield, soil organic carbon and soil enzyme activity as three soil health indicators among the top five keywords. The term "yield" ranked second in the list with 100 documents (10.2% of all articles). Therefore, the yield is considered as part of the soil ecosystem service and it is an important indicator of soil health, as it directly relates to the capacity of the soil to sustain agricultural production and provide food for human populations [36,37]. It occurred for the first time with one document in the second year period (2001–2005). However, it has increased to 44 papers since 2016. Moreover, both soil organic carbon and soil enzyme activity provide information about nutrient cycling, soil structure, water retention, carbon sequestration, and the overall biological activity in the soil [38–40].

Monitoring these indicators helps assess soil health, make informed management decisions, and promote sustainable agricultural practices. Soil organic carbon first appeared in 1996–2000 with one document, while soil enzyme activity first appeared in 2001–2005 with one document, but by the fifth analysed period (2016–2021), they had increased to 60 and 55 documents, respectively.

According to our analysis, the half of the top 20 author keywords is associated with soil management (Table 9). This significant contribution of these 10 soil managements highlights the scientific community's interest in analysing the impact of these soil management strategies on soil health. Our findings indicate that the term "manure" as cropping practice is ranked fourth in terms of relevance, with a total of 75 documents (7.6%) involving this keyword, and it appears for the first time with 1 document in the first 5-year period (1996–2000), until it reaches 43 papers since 2016.

**Table 9.** Main keywords on Soil health (1996–2021).

| Keyword | 1996–2021 | | 1996–2000 | | 2001–2005 | | 2006–2010 | | 2011–2015 | | 2016–2021 | |
|---|---|---|---|---|---|---|---|---|---|---|---|---|
| | A | (%) | A | (%) | A | (%) | A | (%) | A | (%) | A | (%) |
| soil health | 240 | 24.4% | 5 | 71.4% | 5 | 19.2% | 15 | 28.3% | 46 | 28.4% | 169 | 22.9% |
| yield | 100 | 10.2% | 0 | 0.0% | 1 | 3.8% | 7 | 13.2% | 48 | 29.6% | 44 | 6.0% |
| soil organic carbon | 85 | 8.6% | 1 | 14.3% | 5 | 19.2% | 3 | 5.7% | 16 | 9.9% | 60 | 8.1% |
| manure | 75 | 7.6% | 1 | 14.3% | 1 | 3.8% | 3 | 5.7% | 27 | 16.7% | 43 | 5.8% |
| soil enzyme activity | 68 | 6.9% | 0 | 0.0% | 1 | 3.8% | 6 | 11.3% | 6 | 3.7% | 55 | 7.5% |
| soil microbial biomass | 63 | 6.4% | 0 | 0.0% | 4 | 15.4% | 8 | 15.1% | 11 | 6.8% | 40 | 5.4% |
| conservation agriculture | 60 | 6.1% | 0 | 0.0% | 1 | 3.8% | 2 | 3.8% | 9 | 5.6% | 48 | 6.5% |
| sustainability | 59 | 6.0% | 2 | 28.6% | 1 | 3.8% | 5 | 9.4% | 10 | 6.2% | 41 | 5.6% |
| compost | 56 | 5.7% | 1 | 14.3% | 2 | 7.7% | 4 | 7.5% | 11 | 6.8% | 38 | 5.2% |
| cover crop | 56 | 5.7% | 1 | 14.3% | 0 | 0,0% | 1 | 1.9% | 5 | 3.1% | 49 | 6.6% |
| soil quality | 56 | 5.7% | 0 | 0.0% | 2 | 7.7% | 8 | 15.1% | 12 | 7.4% | 34 | 4.6% |
| no tillage | 55 | 5.6% | 0 | 0.0% | 2 | 7.7% | 1 | 1.9% | 8 | 4.9% | 44 | 6.0% |
| soil microbial community | 54 | 5.5% | 0 | 0.0% | 0 | 0.0% | 3 | 5.7% | 2 | 1.2% | 49 | 6.6% |
| fertilization | 53 | 5.4% | 2 | 28.6% | 0 | 0.0% | 4 | 7.5% | 16 | 9.9% | 31 | 4.2% |
| soil properties | 52 | 5.3% | 0 | 0.0% | 1 | 3.8% | 5 | 9.4% | 8 | 4.9% | 38 | 5.2% |
| crop residue management | 51 | 5.2% | 1 | 14.3% | 2 | 7.7% | 4 | 7.5% | 9 | 5.6% | 35 | 4.7% |
| tillage | 50 | 5.1% | 1 | 14.3% | 0 | 0.0% | 8 | 15.1% | 12 | 7.4% | 29 | 3.9% |
| integrated nutrient management | 40 | 4.1% | 0 | 0.0% | 0 | 0.0% | 1 | 1.9% | 11 | 6.8% | 28 | 3.8% |
| wheat | 37 | 3.8% | 0 | 0.0% | 3 | 11.5% | 4 | 7.5% | 10 | 6.2% | 20 | 2.7% |
| crop rotation | 35 | 3.6% | 1 | 14.3% | 2 | 7.7% | 6 | 11.3% | 7 | 4.3% | 19 | 2.6% |

A: number of articles; %: Percentages of articles in which it appears.

Table 9 shows another interesting item. The time evolution of SH papers moved from an early emphasis (e.g., years 1996–2000) on farm management (e.g., fertilization, tillage, rotation) towards a more recent (years 2016–2021) biochemistry and microbiological approaches. This trend depicts the progress in soil biology towards estimating soil health.

Figure 7a shows the keywords co-occurrence network analysis of 100 most relevant keywords which appeared at least two times in 985 articles, which includes 98 nodes, 1055 links, and 2283 total link strength. Each node in the network represents a keyword, and the size of the node reflects the number of times the keyword appeared. According to Yang and Zhuang [41], the presence of keywords with higher occurrences within specific time periods suggests that the corresponding topics are of significant interest and focus during those periods. The network is organized into four clusters of keywords that share similar topics, with the red cluster relating to management strategies and soil health indicators. Because it is a well-developed and important theme with 57 keywords that focuses on various aspects of soil management and soil indicators, this cluster is known as the "motor themes". On the other hand, the green cluster is in second spot in terms of keyword density (26 keywords), and it includes topics linked to the crop yield and nutrient management. This cluster focuses spatially to enhancing crop productivity and soil health using different sustainable crop nutrition strategies such as green manure, compost, bio-fertilizer, and integrated nutrient management. This particular cluster is categorized as fundamental and pertinent across various research fields. The third cluster, represented by the colour blue and containing eleven keywords, deals with soil health and biological activity. This cluster examines the effects of wastewater irrigation and heavy metal contamination on soil biological activity. This group reflects an isolated theme with limited relevance to our research topic. Finally, the yellow cluster can be identified by means of four specific keywords. This particular cluster exhibits a correlation between soil health and greenhouse gas emissions. This cluster was viewed as a marginal and underdeveloped subject.

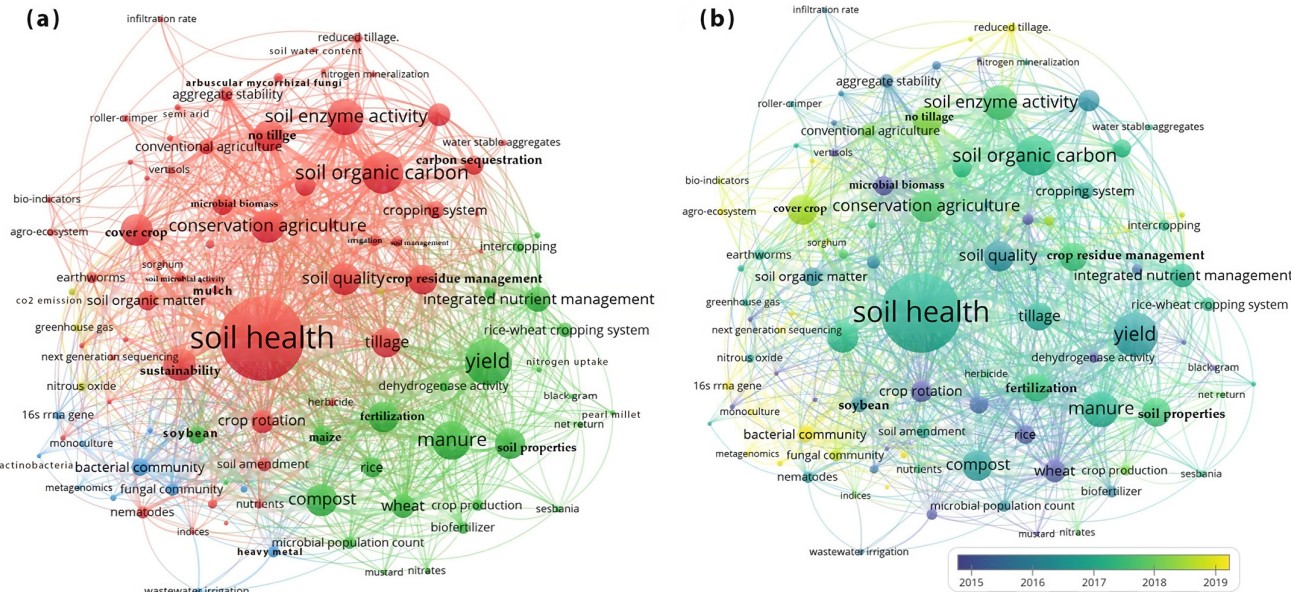

**Figure 7.** (**a**) Co-occurrence network visualization of 100 most relevant keywords which appeared at least 2 times in 985 articles. Colour: represents a cluster of keywords (please see the text for more details); Nodes: represent keywords (node size based on number of occurrences); (**b**) Evolution of the authors' keywords on research topic from 2015 to 2019. Source: VOSviewer (clusters resolution 0.8; minimum cluster size 1 and no merge small clusters).

From the time frame 1996–2021 here in Figure 7b, we report the focus on the evolution of the authors' keywords in the co-occurrence network between 2015 to 2019. Our analysis revealed that keywords related to the impact of soil management such as crop rotation, irrigation, and heavy metal contamination on agricultural soils appeared earlier (before 2015), while keywords related to the impact of cropping practices such as reduced tillage on biochemistry and microbiological soil activities appeared later (in 2019).

The keywords co-occurrence analysis (1996–2021), examining the interplay between agronomic practices and soil health, reveals a landscape rich with insights yet dotted with significant research gaps. The overall scenario (e.g., Figure 7) depicts a strong fragmentation of approaches with many separated analyses about the interaction between soil health and management practices. Basically, a plethora of diverse approaches with limited to almost none attempt of synthesis. In more detail, while the literature showcases strides in understanding short-term impacts of individual practices, it falls short in assessing their long-term implications. A notable opportunity lies in unravelling the effects of holistic approaches that combine various practices, as well as in exploring understudied crops and regions. Amid the intricate tapestry of soil health, microbial interactions emerge as a focal point deserving deeper exploration, promising novel pathways to enhance agricultural sustainability. There is an evident lack on the evaluation of ecosystem services which are becoming crucial in soil health policy across the globe (e.g., EU, USA).

Moreover, the analysis hints at the need to broaden the scope beyond scientific aspects, venturing into economic feasibility, social acceptance, and adaptation challenges. As climate change looms large, the resilience of soil health warrants a spotlight, urging researchers to investigate practices that fortify soil against evolving climatic pressures. The synthesis of the existing literature through meta-analyses is a missing piece that could provide a comprehensive overview, complementing field-scale studies that bridge the gap between controlled experiments and real-world complexities. Collaborative efforts across disciplines hold the potential to yield more holistic insights. Embracing technological and scientific advancements, such as precision agriculture, remote sensing and spatially explicit decision support systems, could further propel the understanding of agronomic impacts on soil health. In conclusion, these research gaps present an invigorating pathway forward,

beckoning scientists, practitioners, and policymakers to collectively foster a more resilient and sustainable agricultural future.

## 4. Conclusions

This bibliometric analysis aims to provide a comprehensive review of the research topic concerning the effects of agronomic practices on soil health. This topic has gained significant importance in both political and scientific areas since the beginning of the second millennium. Based on our data, scholarly production on this research topic has increased significantly over the previous 26 years, with a spike occurring after 2013. These findings supported those of Liu et al. [42], who discovered that since 2013, researchers have become increasingly conscious of the significance of soil health research, and the number of publications published has significantly grown. According to our bibliometric analysis, it can be determined that the journal Applied Soil Ecology and Soil and Tillage Research exhibited the highest level of productivity. The majority of research institutions that investigate our chosen research topic are situated in the United States and India. These nations are consistently ranked among of the most international cooperative in their respective fields. However, when it comes to research institutions, the United States has not achieved the desired level of bilateral cooperation, unlike certain Indian research institutions where international institutional cooperation reached 41 percent. According to the keywords co-occurrence analysis, our research topic is gradually shifting from the effects of soil management strategies, such as nutrient management, on soil indicators, particularly soil organic carbon and yield to the study of the effects of cropping practices on soil biology and biochemistry and greenhouse gas emissions. The last two themes only emerged in the last decade and were seen to be marginal and underdeveloped themes; however, they might provide a promising topic for further study. Overall, our analysis depicts a very large number of soil health research work lacking more integrated and holistic approaches especially in view of analysing the connection between soil health and soil-based ecosystem services.

This study has one limitation, which is that our analysis included only articles from Scopus, and therefore our research cannot cover the entire literature on our research topic. However, the data presented in this study still hold significant potential for understanding the evolving patterns before and after the increase in research on the topic. Finally, this study emphasizes the importance of incorporating additional novel approaches, such as systematic reviews, in order to acquire a more comprehensive understanding of the effects of soil management practices on soil health, and to establish a framework for future research.

**Author Contributions:** Conceptualization, M.H.S. and F.T.; methodology, M.H.S. and F.T; software, M.H.S.; investigation, M.H.S.; data curation, M.H.S.; writing—original draft preparation, M.H.S. and F.T.; writing—review and editing, M.H.S. and F.T.; visualization, M.H.S.; supervision, F.T. All authors have read and agreed to the published version of the manuscript.

**Funding:** This research was funded by European Union's Horizon Europe research and innovation program, under Grant Agreement: 101091010, Project BENCHMARKS.

**Institutional Review Board Statement:** Not applicable.

**Informed Consent Statement:** Not applicable.

**Data Availability Statement:** Not applicable.

**Conflicts of Interest:** The authors declare no conflict of interest.

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
