# Peer review of "Research Evolution on the Impact of Agronomic Practices on Soil Health from 1996 to 2021: A Bibliometric Analysis"

_soilsystems, doi:10.3390/soilsystems7030078_

Round 1

Reviewer 1 Report

The manuscript entitled “Research evolution on the impact of agronomic practices on soil health from 1996 to 2021: A bibliometric analysis” provides an interesting study for the scientific community. However, the manuscript needs to be improved in order to be suitable for publication. The author has written the results with novel information, contribute to the advancement of understanding and are pragmatic. Discussion of the manuscript is well written. In overall manuscript I will suggest that please include

1.       Specific, detailed comments regarding the originality, scientific quality.

2.       Check the needs for tables and figures and the adequacy of the references.

3.       Check the spellings.

The present work is novel and well written.

Author Response

Dear Reviewer,

Thank you indeed for reviewing the manuscript.

Here you found my answers to your valuable comments.

My coauthor and I hope the manuscript amelioration is sufficient for acceptance.

Kind regards.

Dear Reviewer, please, note each of your comments precedes our answer.

  1. Specific, detailed comments regarding the originality, scientific quality.

Answer 1: We didn’t understand this comment by reviewer. Does he want that we comment about the scientific quality of our paper?

  1. Check the needs for tables and figures and the adequacy of the references.

Answer 2: We made full check for tables and figures and the adequacy of the references

  1. Check the spellings.

Answer 3: As suggested by the reviewer the manuscript was grammatically improved and check it by native English mother tongue expert.

Reviewer 2 Report

After a careful reading of the manuscript “Research evolution on the impact of agronomic practices on soil health from 1996 to 2021: A bibliometric analysis”, I can make the following general considerations:

It is a bibliometric analysis of the evolution of research on agronomic practices and their effects on soil health. The manuscript is very well written, it is understandable and interesting to read, the sections are coherent, the explanation sequence of the works carried out is correct, the tables and graphs presented are explanatory and of sufficient quality and, finally, the conclusions are reasonable and agree with the data presented.

Having said the above considerations, I must admit that I am not a specialist in bibliometrics and therefore I am not thoroughly familiar with these studies and with the tools used, particularly the VOSviewer. In any case, as a specialist in soils and agronomy, I can report that the article is interesting to read, although it is the editor who must decide if the manuscript is within the focus of the magazine.

Particular considerations:

Line 153.- In this table in which the eligibility criteria are described, it appears in “Intervention” that all management practices in agronomy have been admitted except for some. However, in the keywords that appear in Table 8, equivalent concepts are considered: for example, "Land Cover" which has theoretically been excluded, "Cover Crop" appears in the keywords admitted and studied. Perhaps it would be interesting to clarify this exclusion and then, the use of similar keywords.

Line 172.- The data analysis in general corresponds to a period of 26 years from 1996 to 2021. It would be interesting to note that, in the last part, the analysis of the keywords is only carried out for a period of 5 years.

Line 211.- It would be interesting to point out in this figure that the exponential curve is a theoretical calculation and it is only used to compare the fit with the real data.

Line 316.- The analysis of authors appears clearly conditioned by the large number of Asian and some North American journals. The 10 most relevant authors appear to be of Asian origin. Probably the numbers and the bibliometric analysis indicate this, but we must not lose sight of the fact that their publications are produced in local journals and with little international impact, as reflected in the following tables. Perhaps it would be interesting to make this consideration in the text.

Line 559, and figure 7.- very interesting graph, perhaps a bit cumbersome due to the large number of keywords considered. However, it is very interesting from my point of view, even though its analysis is only carried out in the last 5 years of the study. This limitation should be made very clear when drawing conclusions that are interesting in any case.

Author Response

Dear Reviewer,

Thank you indeed for reviewing the manuscript.

Here you found my answers to your valuable comments.

My coauthor and I hope the manuscript amelioration is sufficient for acceptance.

Kind regards.

Dear Reviewer, please, note each of your comments precedes our answer.

Particular considerations:

Line 153.- In this table in which the eligibility criteria are described, it appears in “Intervention” that all management practices in agronomy have been admitted except for some. However, in the keywords that appear in Table 8, equivalent concepts are considered: for example, "Land Cover" which has theoretically been excluded, "Cover Crop" appears in the keywords admitted and studied. Perhaps it would be interesting to clarify this exclusion and then, the use of similar keywords.

Answer: We would like to clarify the distinction between "Land cover" and "cover crops" based on your feedback. These two terms have distinct meanings in the context of agriculture and land use. While land cover refers to the physical and biological cover of the land surface (basically it refers to very aggregated land use classes such as forest, grassland, urban areas, and water bodies), cover crops are non-commercial species planted in between cash crops in a crop rotation to improve soil health and prevent soil erosion.

The exclusion of the "land cover" criteria has been produced after a preliminary analysis which convinced us that the strong aggregation of land use classes typically occurring when using the "crop cover" criteria would be misleading.

Line 172.- The data analysis in general corresponds to a period of 26 years from 1996 to 2021. It would be interesting to note that, in the last part, the analysis of the keywords is only carried out for a period of 5 years.

Answer: We fixed the text of the manuscript on the base of your doubt. In fact, It appears there might be some confusion regarding Figure 7b, which depicts the Evolution of the authors' keywords on research topic from 2015 to 2019. Your insight is greatly appreciated, and I would like to clarify that the analysis in our study encompasses the entire 26-year duration. The software, as you pointed out, emphasizes the significance of the 2015-2019 timeframe in terms of research progression. This specific period indeed showcases a notable shift in the research landscape, as outlined in Figure 7b. However, our analysis provides a comprehensive overview of the entire 26-year span, thereby offering a complete perspective on the evolution of the research topic.

Line 211.- It would be interesting to point out in this figure that the exponential curve is a theoretical calculation and it is only used to compare the fit with the real data.

Answer: As suggested by the reviewer we fixed the text in the figure 2.

Line 316.- The analysis of authors appears clearly conditioned by the large number of Asian and some North American journals. The 10 most relevant authors appear to be of Asian origin. Probably the numbers and the bibliometric analysis indicate this, but we must not lose sight of the fact that their publications are produced in local journals and with little international impact, as reflected in the following tables. Perhaps it would be interesting to make this consideration in the text.

Answer: As suggested by the reviewer we added this paragraph at the line 341:” The prominence of Asian authors among the top 10 most relevant contributors is evident, a trend that is likely reflected in the numbers and supported by bibliometric analysis. However, it's crucial to maintain perspective regarding their publications, primarily confined to local journals and demonstrating limited international impact. This serves as a reminder that while numerical metrics highlight certain patterns, considering the qualitative aspects of influence is equally vital.

Line 559, and figure 7.- very interesting graph, perhaps a bit cumbersome due to the large number of keywords considered. However, it is very interesting from my point of view, even though its analysis is only carried out in the last 5 years of the study. This limitation should be made very clear when drawing conclusions that are interesting in any case.

Answer: We fixed the text of the manuscript on the base of the reviewer comment. In fact, It appears there might be some confusion regarding Figure 7b, which depicts the Evolution of the authors' keywords on research topic from 2015 to 2019. Your insight is greatly appreciated, and I would like to clarify that the analysis in our study encompasses the entire 26-year duration. The VOSviewer software, as you pointed out, emphasizes the significance of the 2015-2019 timeframe in terms of research progression. This specific period indeed showcases a notable shift in the research landscape, as outlined in Figure 7b. However, our analysis provides a comprehensive overview of the entire 26-year span (e.g., Figure 7 a), thereby offering a complete perspective on the evolution of the research topic.

Reviewer 3 Report

Dear authors,

First of all, congratulate you for the work carried out in this article, being of interest in terms of agronomic practices on soil health.

Material and method are well planned and detailed information is given.

However, under the heading Subject area and journals

   -Agricultural and Biological Sciences,

- Environmental Science category.

  - Earth and Planetary Sciences

- Immunology and Microbiology

  - Biochemistry, Genetics, and Molecular

- Biology

 These studies with sub-headings could be given. It would be more understandable if we were groups in these sub-headings.

Best regards,

Author Response

Dear Reviewer,

Thank you indeed for reviewing the manuscript.

You found here my answers to your valuable comments.

My coauthor and I hope the manuscript amelioration is sufficient for acceptance.

Kind regards.

Dear Reviewer, please, note each of your comments precedes our answer.

However, under the heading Subject area and journals

   -Agricultural and Biological Sciences,

 - Environmental Science category.

  - Earth and Planetary Sciences

- Immunology and Microbiology

  - Biochemistry, Genetics, and Molecular

- Biology

 These studies with sub-headings could be given. It would be more understandable if we were groups in these sub-headings.

Answer: If we have properly understood the reviewer, he would like that we reorganized the papers and journals in subject areas. We doubt that this internal categorization made by Scopus for internal proposes could be beneficial for the reader

Reviewer 4 Report

1.      Too many short paragraphs existed in the introduction or results and discussion.

2.      The last paragraph in the introduction was not needed.

3.      The objectives of this study can be presented in the last part of the introduction.

4.      Try to beautify the figures further.

5.      The horizontal line in Fig.2 was needed, but the fitted line in Fig.2 was not.

6.      Some colors in Fig.4 were not easy to distinguish.

7.      The discussion should be strengthened further.

8.      No citation was needed in the discussion.

It was well written in English

Author Response

Dear Reviewer,

Thank you indeed for reviewing the manuscript.

You found here my answers to your valuable comments.

My coauthor and I hope the manuscript amelioration is sufficient for acceptance.

Kind regards.

Dear Reviewer, please, note each of your comments precedes our answer.

 1-Too many short paragraphs existed in the introduction or results and discussion.

Answer 1: Most reviewers congratulate us for the English. We would prefer to keep the sentences as they are.

  1. The last paragraph in the introduction was not needed.

Answer 2:  We deleted the last sentence but - for the good sake of readability - we have kept the explanation concerning how the paper is structured.

  1. The objectives of this study can be presented in the last part of the introduction.

Answer 3: We reworded the sentence at line 105 to better clarify our aims

  1. Try to beautify the figures further.

Answer 4: As suggested by the reviewer we improved the figure 4 and 5.

  1. The horizontal line in Fig.2 was needed, but the fitted line in Fig.2 was not.

Answer 5: We have already fulfilled the request on this figure by reviewer 2

  1. Some colors in Fig.4 were not easy to distinguish.

Answer 6: As suggested by the reviewer we improved the figure 4  

  1. The discussion should be strengthened further.

Answer 7: We have reinforced the discussion emphasizing in our view

  1. No citation was needed in the discussion.

Answer 8: we disagree since in the discussion is very much required to compare any research work with those produced by other authors

Reviewer 5 Report

Dear authors,

I would like to extend my warmest congratulations to you for your sincere efforts in bringing out this manuscript. I must say that the manuscript is very well prepared, and I appreciate the elaborate description of the introduction, material, and methods. The results are also properly described and well discussed, and it's great that you have honestly pointed out the limitations of this study.

However, I have one observation regarding the use of bibliometric analysis. I wonder how the scientific community would benefit from this study. It would have been excellent if you had highlighted the research gaps that emerged from this bibliometric analysis and deserve the attention of the scientific community. Also, some parts of the study, such as the generalized statements from line no. 620 to 623, could have been more specific. I would also suggest to use the word “study” instead of “research” in line no. 620. Additionally, in the abstract at line no. 26-27, you mentioned sustainable management practices, but what does it mean?

Lastly, I noticed some typological mistakes, such as in line no. 330, where the word "are" should be removed before the word "published." Similarly, at line no. 472, please replace the word "son" with "on."

Author Response

Dear Reviewer,

Thank you indeed for reviewing the manuscript.

You found here my answers to your valuable comments.

My coauthor and I hope the manuscript amelioration is sufficient for acceptance.

Kind regards.

Dear Reviewer, please, note each of your comments precedes our answer.

However, I have one observation regarding the use of bibliometric analysis. I wonder how the scientific community would benefit from this study. It would have been excellent if you had highlighted the research gaps that emerged from this bibliometric analysis and deserve the attention of the scientific community. Also, some parts of the study, such as the generalized statements from line no. 620 to 623, could have been more specific. I would also suggest to use the word “study” instead of “research” in line no. 620. Additionally, in the abstract at line no. 26-27, you mentioned sustainable management practices, but what does it mean?

Lastly, I noticed some typological mistakes, such as in line no. 330, where the word "are" should be removed before the word "published." Similarly, at line no. 472, please replace the word "son" with "on."

Answer:

  • We have reinforced the discussion emphasizing in our view please see the line 605 in the manuscript.
  • our text line 620-623 are required to highlight the importance of systematic reviews as a comprehensive and structured approach to summarizing existing research on soil science frontier such as soil health analysis.
  • The sustainable management practices are those enabling soils to continue to produce ecosystem services without any soil degradation.
  • The manuscript was grammatically improved and check it by native English mother tongue expert.

Reviewer 6 Report

This submission deals with soil sciences and particularly with publications dealing with the health of soils.

In my opinion, it is of sufficient interest and acceptable for publication. Yet, the use of terms and phrases is not always precise (or correct) hence some changes should be made.

Precision needed

I do not understand the title. I think writing “Evolution of the impact of agronomic practices on soil health: A bibliometric analysis over the period 1996-2021. “would be better.

Line 20-21 The USA and India are not research institutes (as written) but countries.

Line 47 Do ecosystems require to be measurable?  I do not understand this (English!)

Line 61 “the huge divide between us scientists, experts and citizens”. Which divide? Please clarify. 

Line 88 What is a “lighthouse”? For me, it is “A lighthouse is a tower, building, or other type of physical structure designed to emit light from a system of lamps and lenses and to serve as a beacon.” But obviously the word has been used with a different meaning.

Line 135. All data come from Scopus, but here the Web of Science is used. Why? It would be logical to use Scopus.

Line 187-190. Repetition of things that are already mentioned. Delete or shorten.

Table 2. Explain the meaning of TC. Is it the total number of citations from the moment of publication till now (or till 2022); or is it the total number of citations received during the period indicated in the first column?

Figure 2. Redo this figure.  It makes no sense to fit a curve when e.g., the void between [2001-2005] and [2006-2010] represents no data.

Line 233. For me writing has increased 143 times is easier to understand than an expression in percentages. Note: it is 143 and not 142 (or did I miscalculate?).

Line 238. Also here. Please write “has increased more than 13 times”

Line 329-331. I would remove this sentence. In my opinion, it is not essential.

Linguistic corrections

Line 17 Replace ‘under” with “in”

Line 22 The part “and neither do scholars” makes no sense (linguistically). Please rephrase.

Line 32 Replace “herewith after named as SH” with “from now on abbreviated as SH”

Line 36 Replace “initiative” with “initiatives”

Line 48 Replace “soils health with” with “Soil health and”

Line 68 End the sentence after [8]. Then begin a new sentence “It is not …”

Line 57 “we agreed”. This gives the impression that at some point you agreed, but not anymore. It would be better to replace “we agreed” with “we agree”.

Line 60 Delete the word “it”

Line 70 Replace “In addition” with “moreover”

Line 73 Replace “practice” with “practices”

Line 76. End sentence after (42 countries). Then write “They found 42 indicators … “

Line 110 Replace “analyzing” with “analysis”

Line 239 Replace “on” with “in”

Line 307. Write “the 20 most prolific journals”

Line 307 Write “but the five articles”

Line 583 Replace “reflected” with “reflects”

Line 606 Write “… ranked among the most …”

A general remark

When I learned English the word “last” meant that nothing came later, as in “the last day of the month”. The word “latest” meant “most recent”.  Nowadays many people do not make this distinction anymore and always use “last”, so this use is considered to be correct English. Yet, it is a pity that this useful distinction is not made anymore. In your text, the word ‘last’ usually means ‘latest” as in the latest 26 years (line 18). May I suggest checking your word use and change when necessary? It really would make the text more precise (and easier to understand).

Author Response

Dear Reviewer,

Thank you indeed for reviewing the manuscript.

You found heremy answers to your valuable comments.

My coauthor and I hope the manuscript amelioration is sufficient for acceptance.

Kind regards.

Dear Reviewer, please, note each of your comments precedes our answer.

Precision needed

I do not understand the title. I think writing “Evolution of the impact of agronomic practices on soil health: A bibliometric analysis over the period 1996-2021. “would be better.

Answer : We prefer to leave our title since we did not analyse the evolution of the impact of agronomic practices but rather the evolution of soil health research.

Line 20-21 The USA and India are not research institutes (as written) but countries.

Answer:  We improved this sentence please see the line 20-21 of the manuscript

Line 47 Do ecosystems require to be measurable?  I do not understand this (English!)

Answer:  The quantification of ecosystem services is required by important soil policies (e.g. the EU New Soil Strategy)

Line 61 “the huge divide between us scientists, experts and citizens”. Which divide? Please clarify. 

AnswerThis is a well-known problem between policy and science. Here as example, we provide a couple of examples:

1-https://implementationscience.biomedcentral.com/articles/10.1186/s13012-016-0377-7

2-https://www.sbu.se/en/publications/medical-and-science-newsletter/bridging-the-sciencepolicy-gap/#:~:text=Many%20researchers%20have%20described%20the,on%20values%20and%20available%20resources

Line 88 What is a “lighthouse”? For me, it is “A lighthouse is a tower, building, or other type of physical structure designed to emit light from a system of lamps and lenses and to serve as a beacon.” But obviously the word has been used with a different meaning.

Answer:  We fixed the problem in the text. Anyway, we clarify this issue also here:

What is a Living Lab?

Living Labs are places to experiment on the ground. They are collaborative initiatives between multiple partners and diverse actors, like researchers, farmers, foresters, spatial planners, land managers, and citizens who come together to co-create innovation for jointly agreed objectives. Living Labs are established on territorial, landscape, or regional scale. They coordinate experiments on multiple sites such as farms, forests, urban or industrial areas.

What is a Lighthouse?

Lighthouses are single sites, like farms or parks, where scientifically proven good practices and solutions are demonstrated. Lighthouses can also be single sites that are part of the Living Labs.

They are places for mutual exchange and peer-to-peer learning. Good practices are further tested under real life conditions to inspire other practitioners to move towards sustainable soil and land management.

Line 135. All data come from Scopus, but here the Web of Science is used. Why? It would be logical to use Scopus.

Answer:  The Scopus database does not offer distinct subject categories for agronomy and soil science; rather, they are encompassed under the umbrella of "Agricultural and Biological Sciences." This categorization presents a limitation in terms of classifying specific subfields.

Line 187-190. Repetition of things that are already mentioned. Delete or shorten.

Answer: We improved this sentence please see the line 187 of the manuscript.

Table 2. Explain the meaning of TC. Is it the total number of citations from the moment of publication till now (or till 2022); or is it the total number of citations received during the period indicated in the first column?

Answer: Total number of citations from the moment of publication till Jun 25, 2023.

Figure 2. Redo this figure.  It makes no sense to fit a curve when e.g., the void between [2001-2005] and [2006-2010] represents no data.

Answer: we followed the suggest of the reviewer 2

Line 233. For me writing has increased 143 times is easier to understand than an expression in percentages. Note: it is 143 and not 142 (or did I miscalculate?).

Answer: As suggested by the reviewer we improved the text line 233

Line 238. Also here. Please write “has increased more than 13 times”

 Answer: As suggested by the reviewer we improved the text line 238

Line 329-331. I would remove this sentence. In my opinion, it is not essential.

Answer: As suggested by the reviewer we deleted this sentence.

Comments on the Quality of English Language

Linguistic corrections

Line 17 Replace ‘under” with “in”

Answer: it was done.

Line 22 The part “and neither do scholars” makes no sense (linguistically). Please rephrase.

Answer: We improved the sentence. Please the line 22 in the manuscript.

Line 32 Replace “herewith after named as SH” with “from now on abbreviated as SH”

Answer: it was done.

Line 36 Replace “initiative” with “initiatives”

Answer: it was done.

Line 48 Replace “soils health with” with “Soil health and”

Answer: it was done.

Line 68 End the sentence after [8]. Then begin a new sentence “It is not …”

Answer: it was done.

Line 57 “we agreed”. This gives the impression that at some point you agreed, but not anymore. It would be better to replace “we agreed” with “we agree”.

Answer: it was done.

Line 60 Delete the word “it”

Answer: it was done.

Line 70 Replace “In addition” with “moreover”

Answer: it was done.

Line 73 Replace “practice” with “practices”

Answer: it was done.

Line 76. End sentence after (42 countries). Then write “They found 42 indicators … “

Answer: it was done.

Line 110 Replace “analyzing” with “analysis”

Answer: it was done.

Line 239 Replace “on” with “in”

Answer: it was done.

Line 307. Write “the 20 most prolific journals”

Answer: it was done.

Line 307 Write “but the five articles”

Answer: it was done.

Line 583 Replace “reflected” with “reflects”

Answer: it was done.

Line 606 Write “… ranked among the most …”

Answer: it was done.

Round 2

Reviewer 4 Report

All of my comments were well responded to. 

Another comment was given due to the mistake of the 8th comment some days ago.

  1. No citation was needed in the conclusions.
  1.